DOI: 10.1038/s41467-018-04188-7　　**OPEN**

# Hematopoietic stem cells can differentiate into restricted myeloid progenitors before cell division in mice

Tatyana Grinenko[1], Anne Eugster[2], Lars Thielecke[3], Beáta Ramasz[1], Anja Krüger[1], Sevina Dietz[2], Ingmar Glauche [3], Alexander Gerbaulet[4], Malte von Bonin [5,6,7], Onur Basak[8,9,10], Hans Clevers[8,9,11], Triantafyllos Chavakis[1,2] & Ben Wielockx [1,2]

Hematopoietic stem cells (HSCs) continuously replenish all blood cell types through a series of differentiation steps and repeated cell divisions that involve the generation of lineage-committed progenitors. However, whether cell division in HSCs precedes differentiation is unclear. To this end, we used an HSC cell-tracing approach and Ki67$^{RFP}$ knock-in mice, in a non-conditioned transplantation model, to assess divisional history, cell cycle progression, and differentiation of adult HSCs. Our results reveal that HSCs are able to differentiate into restricted progenitors, especially common myeloid, megakaryocyte-erythroid and pre-megakaryocyte progenitors, without undergoing cell division and even before entering the S phase of the cell cycle. Additionally, the phenotype of the undivided but differentiated progenitors correlated with the expression of lineage-specific genes and loss of multipotency. Thus HSC fate decisions can be uncoupled from physical cell division. These results facilitate a better understanding of the mechanisms that control fate decisions in hematopoietic cells.

[1] Department of Clinical Pathobiochemistry, Institute for Clinical Chemistry and Laboratory Medicine, Technische Universität Dresden, Fetscherstraße 74, 01307 Dresden, Germany. [2] DFG Research Centre and Cluster of Excellence for Regenerative Therapies Dresden, Technische Universität Dresden, Fetscherstraße 105, 01307 Dresden, Germany. [3] Institute for Medical Informatics and Biometry (IMB), Technische Universität Dresden, Fetscherstraße 74, 01307 Dresden, Germany. [4] Institute for Immunology, Technische Universität Dresden, Fetscherstraße 74, 01307 Dresden, Germany. [5] Medical Clinic and Policlinic I, University Hospital Carl Gustav Carus, Technische Universität Dresden, Fetscherstraße 74, 01307 Dresden, Germany. [6] German Cancer Consortium (DKTK), partner site Dresden, Fetscherstraße 74, 01307 Dresden, Germany. [7] German Cancer Research Center (DKFZ), Im Neuenheimer Feld 280, 69120 Heidelberg, Germany. [8] Hubrecht Institute, Royal Netherlands Academy of Arts and Sciences and University Medical Center Utrecht, Uppsalalaan 8, 3584 CT Utrecht, The Netherlands. [9] Cancer Genomics Netherlands, UMC Utrecht, Heidelberglaan 100, 3584 CX Utrecht, The Netherlands. [10] Department of Translational Neuroscience, Brain Center Rudolf Magnus, University Medical Center Utrecht and Utrecht University, 3584 CG Utrecht, The Netherlands. [11] Princess Máxima Centre, Lundlaan 6, 3584 EA Utrecht, The Netherlands. Correspondence and requests for materials should be addressed to T.G. (email: tatyana.grinenko@uniklinikum-dresden.de) or to B.W. (email: ben.wielockx@tu-dresden.de)

A rare population of hematopoietic stem cells (HSCs) resides at the top of the hematopoietic hierarchy[1]. Although most adult HSCs normally exist in a quiescent or dormant state[2], some of them divide and support the production of all mature blood cell types through multiple intermediate progenitor stages, during steady state, and in response to acute needs[3–5]. These include myeloid progenitors (MPs), encompassing restricted progenitors like common myeloid progenitors (CMPs), granulocyte-macrophage progenitors (GMPs), pre-megakaryocyte-erythroid progenitors (PreMEs), and pre-megakaryocyte progenitors (PreMegs). This classical point of view was questioned in recent studies from two groups showing that HSC populations contain stem-cell-like megakaryocyte progenitors, which under stress conditions such as transplantation into irradiated recipients[6] or after acute inflammation[7] activate a megakaryocyte differentiation program. The commitment process(es) that turns HSCs into mature cells are currently understood to be a sequence (or even a continuum) of decision steps in which the multilineage potential of the cells is sequentially lost[8–10]. Although many of these steps have been investigated in great detail, the entire picture is still repeatedly challenged[6,8,9,11–13]. HSC transition through the multipotent and restricted progenitor stages is also accompanied by intense cell proliferation[3]. However, it is unclear whether each fate decision step is associated with one or more division events or whether cell proliferation and differentiation are independent processes. Further, if differentiation of HSCs does require cell division, the phase of the cell cycle that is particularly important for this process is also currently unknown. The dependence of cell fate decisions on cell cycle progression was so far only shown in vitro for pluripotent embryonic stem cells[14–17]. However, a few reports point toward a functional connection between these two processes in adult stem cells, such as neuronal stem cells[16,18]. With regard to hematopoietic stem and progenitor cells, characterization of the cell cycle itself is currently ongoing[19–22], and an understanding of how HSC fate decisions relate to cell division and cell cycle progression is lacking[19].

Therefore, we used in vivo cell tracing to simultaneously follow the divisional history and the initial differentiation steps of HSCs. Our data reveal that HSCs are able to differentiate into restricted progenitors prior to cell division, most prominently PreMEs and PreMegs, and that this occurs before the cells enter the S phase of the cell cycle. Moreover, our data also demonstrate that the G0/G1 phases are important for fate decision in HSCs to either differentiate or self-renew.

## Results

**HSCs differentiate into MPs without dividing.** To study the initial steps of HSC differentiation in vivo, we sorted Lin⁻ Kit⁺ Sca-1⁺ (LSK) CD48⁻ CD41⁻ CD150⁺ stem cells (Fig. 1a)[1]. CD41⁺ cells were excluded to reduce myeloid-[23] and megakaryocyte-biased HSCs[24–26]. We used the CellTrace Violet dye[27,28] to uniformly label HSCs and track cell division history after transplantation (Fig. 1a). Recently, Shimoto et al. have shown that numerous empty HSC niches are available upon transplantation into non-conditioned recipients, which are located distant from filled niches and available for HSC engraftment and proliferation. Moreover, donor HSCs give rise to all blood cells without any bias[29]. Labeled cells were transplanted into unconditioned recipients to prevent irradiation-induced stress[30–32] (Fig. 1a). Thirty-six hours after transplantation, 30% of the donor cells had downregulated Sca-1 expression (Fig. 1b), one of the principal surface marker for HSCs[33], and changed their phenotype from HSCs to MPs. Importantly, the purification procedure alone did not lead to downregulation of Sca-1 (Supplementary Fig. 1a). A

possible contamination of potential donor MPs was excluded, since transplantation of these progenitors alone did not result in any detectable donor MPs 36 h later (Supplementary Fig. 1b). To further classify these phenotypically restricted MPs in vivo, we used a gating strategy according to Pronk and colleagues[34]. Briefly, PreMEs (Lin⁻ Sca1⁻ Kit⁺ CD41⁻ CD16/32⁻ CD105⁻ CD150⁺), Pre-CFU-E (Lin⁻ Sca1⁻ Kit⁺ CD41⁻ CD16/32⁻ CD105⁺ CD150⁺), CFU-E (Lin⁻ Sca1⁻ Kit⁺ CD41⁻ CD16/32⁻ CD105⁺ CD150⁻), PreMegs (Lin⁻ Sca1⁻ Kit⁺ CD41⁺ CD16/32⁻ CD150⁺), and Pre-GM (Lin⁻ Sca1⁻ Kit⁺ CD41⁻ CD16/32⁻ CD105⁻ CD150⁻) staining was initially confirmed by transplantation of cells into lethally irradiated mice (Supplementary Fig. 2a, b). However, Pre-GMs gave rise to platelets and myeloid and erythroid cells after transplantation and were therefore classified as CMPs. Based on surface staining at 36 h posttransplantation, we subdivided donor MPs into the following restricted progenitors: CMPs, GMPs (Lin⁻ Sca1⁻ Kit⁺ CD41⁻ CD150⁻ CD16/32⁺), PreMEs, and PreMegs (Fig. 1b).

Next, we analyzed the proliferation history of transplanted cells based on dilution of CellTrace Violet dye, whereby intensity of the dye in CD4⁺ CD62L⁺ naive T cells was used as the reference for undivided cells (Supplementary Fig. 1c)[35,36]. This analysis reveals that, at 36 h after HSC transplantation, a majority of LSK cells with the long-term HSC phenotype (LSK CD48⁻ CD150⁺), short-term HSCs (ST-HSCs) (LSK CD48⁻ CD150⁻), multipotent progenitors (MPP2: LSK CD48⁺ CD150⁺ and MPP3/4 LSK CD48⁺ CD150⁻) (Supplementary Fig. 3a)[1] and 50% of the MPs remained undivided (Fig. 1c). Additionally, based on CD41 and CD150 expression, these MPs were predominantly CMPs, PreMEs, and PreMegs (Fig. 1d, e). We also performed an even more stringent gating strategy to avoid overlay between non-divided and divided cells (Supplementary Fig. 1d) but found no difference in the frequency of restricted progenitors, as compared to the previous gating strategy (Supplementary Fig. 1e). To exclude the possibility that HSCs differentiated into MPs without division due to the limited niche space, we used the HSC-CreERT +R26^DTA/DTA mouse line allowing for the inducible depletion of HSCs and transplanted CellTrace dye labeled wild-type HSCs into them[37] (Supplementary Fig. 1f). However, we did not find any difference in the frequency of HSCs differentiated into myeloid-restricted progenitors 36 h after transplantation, compared to controls (Supplementary Fig. 1g, h). Surprisingly, compared to mice not preconditioned with tamoxifen (TAM), we found that donor HSCs in TAM-treated mice displayed enhanced differentiation into GMPs without cell division, suggesting potentially additional stress induced by TAM.

Interestingly, transplantation of MPP2 or MPP3/4 subsets revealed a similar phenomenon as most of the MPs did not divide. Further, while MPP2 cells mostly gave rise to PreMEs and PreMeg cells, MPP3/4 cells differentiated into CMPs and GMPs (Supplementary Fig. 3b–d). Taken together, these results strongly suggest that HSCs/MPPs can give rise to restricted progenitors including CMPs, PreMEs, and PreMegs based on the cell phenotype, without undergoing cell division.

**Undivided differentiated progenitors express lineage genes.** To investigate the molecular differences between undivided HSCs and undivided MPs, we designed a panel of primers to analyze single-cell expression levels of 70 genes including cell cycle genes and those specific for HSCs, myeloid, erythroid, megakaryocyte-erythroid progenitors (MEP), and platelets (Supplementary Table 1)[8,9,11,12,33,38–41]. Essentially, single-cell expression analysis of freshly sorted HSCs, CMPs, PreMEs, and PreMegs showed a clear separation of the cell types whether based on all analyzed

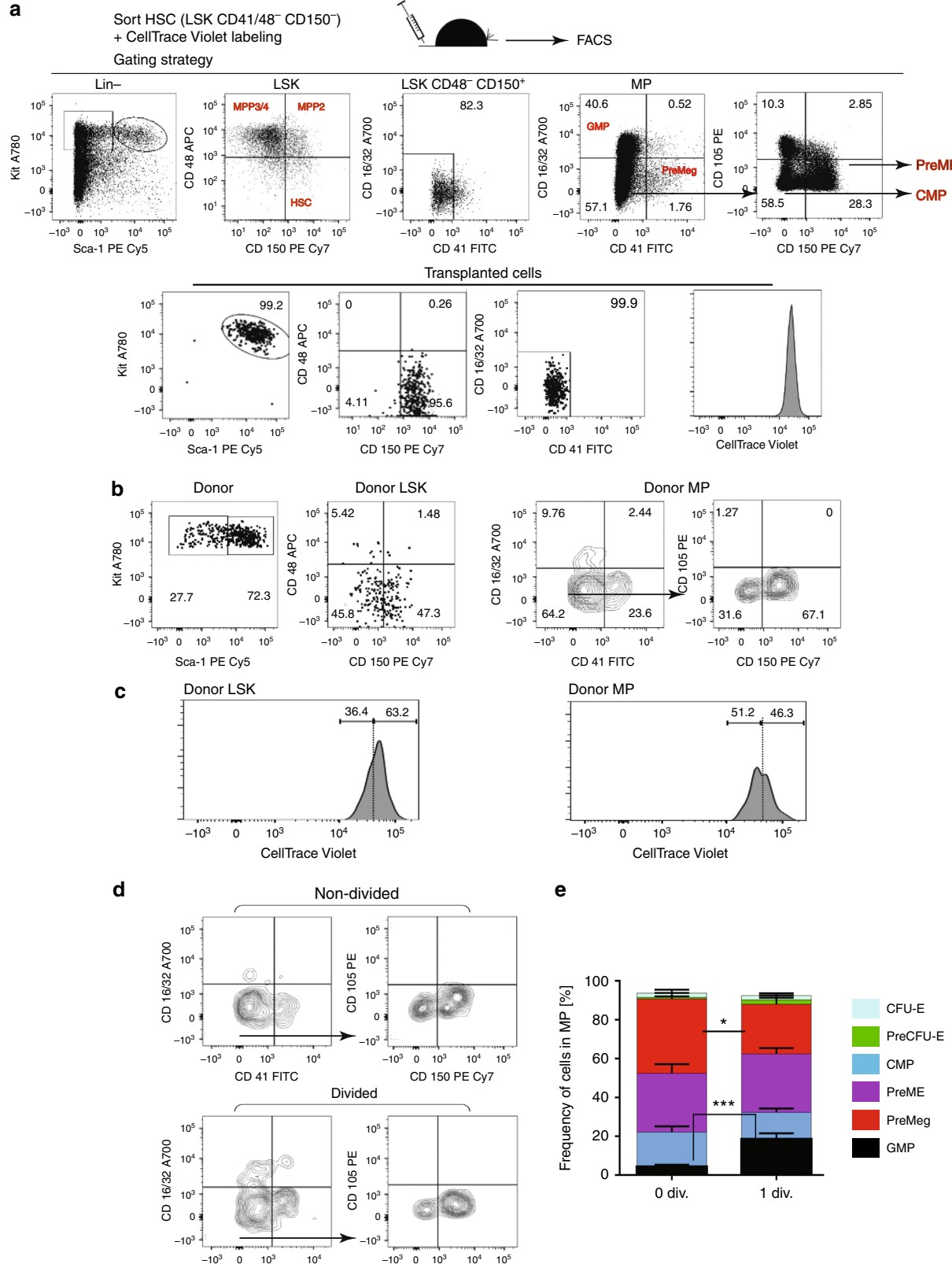

**Fig. 1** Differentiation and division proliferation history of HSCs after transplantation into non-conditioned recipients. **a** HSCs (LSK CD48−CD41−CD150+) were labeled with CellTrace Violet dye and 3600 cells were transplanted into non-conditioned wild-type mice. Purity of transplanted cells was >99% for each experiment. **b** Bone marrow was harvested at 36 h after transplantation and donor cells were analyzed using the indicated gates. **c** Dilution of CellTrace Violet in donor LSK and MPs, 36 h after transplantation. Labeled and transplanted naive CD62L+CD4+ T cells were used as reference for undivided cells. Five hundred donor cells were analyzed from 11 transplanted mice, representative data for 1 out of 13 experiments. **d** Phenotype of undivided and divided donor MPs (n = 11), representative example of 13 independent experiments. **e** Frequency of restricted progenitors in undivided (0 div.) and divided (1 div.) donor MPs, pooled data from 13 independent experiments. Unpaired Student's t-test, data are means +/− s.d., ***P = 0.0002, *P = 0.02

genes or only on selected MEP/platelet genes (Supplementary Fig. 4a–c).

We then isolated undivided donor cells at 36 h after transplantation of LSK CD48⁻ CD41⁻ CD150⁺ cells (Fig. 2a) and retrospectively categorized them on the basis of index sorting data as HSCs (LSK CD48⁻ CD150⁺) or various MP populations (Supplementary Fig. 5a–d). Within these populations, we performed single-cell quantitative PCR (qPCR) on 42 HSCs, 7 CMPs, 15 PreMEs, and 20 PreMegs pooled and obtained from two independent experiments (Fig. 2b). Performing t-distributed

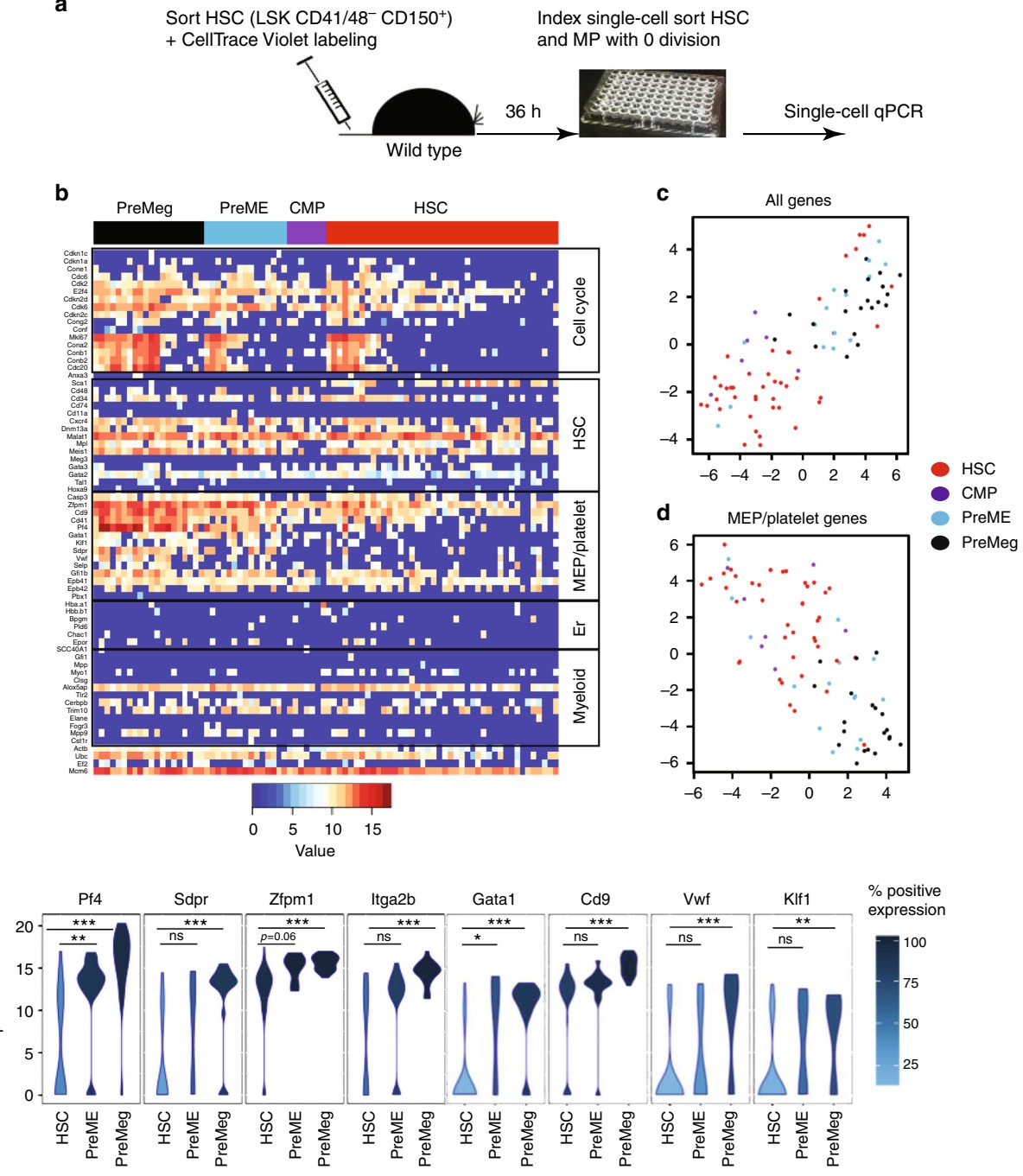

**Fig. 2** Single-cell expression analysis in undivided donor HSCs and MPs. **a** Experimental design. LSK CD48⁻ CD41⁻ CD150⁺ cells were transplanted into non-irradiated recipients, and single, undivided donor Lin⁻Kit⁺ cells were sorted using the index sort approach at 36 h after transplantation. Data from two independent experiments ($n = 12$ mice). Based on index sort data, HSCs were defined as LSK CD48⁻CD150⁺; CMPs as Lin⁻ Kit⁺ Sca-1⁻ CD16/32⁻ CD41⁻ CD150⁻CD105⁻; PreME as Lin⁻Kit⁺ Sca-1⁻ CD16/32⁻ CD41⁻ CD150⁺ CD105⁻; and PreMeg as Lin⁻Kit⁺ Sca-1⁻ CD16/32⁻ CD41⁺ CD150⁺ CD105⁻. All sorted 42 HSC, 7 CMP, 15 PreME, and 20 PreMeg cells were analyzed. **b** Heat map showing gene expression analysis. Each row corresponds to a specific gene, each column corresponds to a specific and individual donor cell, and colors represent the expression levels of individual genes (dCt). **c** t-SNE plot for all the analyzed genes and cells, axes display arbitrary units. **d** t-SNE plot for MEP/platelet genes for all cells, axes display arbitrary units. **e** Violin density plots for the most differently expressed MEP/platelet genes. $y$ Axis represents gene expression. The horizontal width of the plot shows the density of the data along the $y$ axis. Statistical significance was determined using the Hurdle model. *$P < 0.05$; **$P < 0.01$; ***$P < 0.0001$; ns, not significant. Data from two independent experiments, $n = 12$. Exact $P$-value in supplemental Tables S2,3

stochastic neighbor embedding (t-SNE) analysis of the qPCR data revealed separation of HSCs from PreMEs and PreMegs, based on all analyzed genes (Fig. 2c) or the MEP/platelet genes alone (Fig. 2d, Supplementary Table 1). This separation among phenotypically defined populations was also confirmed by a majority of the MEP/platelet-specific genes (Fig. 2e, Supplementary Tables 2, 3, 4, 5) and was similar to that observed before transplantation (Supplementary Fig. 4d). Thus undivided PreME/PreMeg cells obtained after transplantation express genes typically restricted to MEP.

For an in-depth comparative analysis of the transplanted undivided cells (Fig. 2) and non-transplanted cells (Supplementary Fig. 4), we performed t-SNE[42] and hierarchical cluster analysis on gene expression data (Fig. 3a, b, Supplementary Fig. 5e). We wondered whether HSCs and PreMegs truly form distinctive subgroups in terms of their gene expression profile. Therefore, we excluded the intermediate cell differentiation stages (colored in green) and provided the algorithm with a number of expected clusters ($k = 2$). Figure 3b illustrates that not only the visual inspection of the t-SNE visualization but also the $k$-means cluster algorithm is able to distinguish between those two cell types. As expected, while our results reveal a close association between the before- and after-transplantation HSC or PreMeg populations, HSCs and PreMegs themselves form distinct clusters. Therefore, changes in the HSC phenotype before cell division reflect gene expression changes associated with differentiation.

**HSCs differentiate before the S phase of the cell cycle**. While the cell-tracing dye allowed us to follow cell division, it did not give information on cell cycle progression. Therefore, to determine in which phase of the cell cycle HSCs make fate decisions, we scored each cell for its likely cell cycle phase using signatures for G1 and S/G2/M phases[39]. We categorized individual cells in the G0/G1 or the S/G2/M phases (Fig. 4a) based on the average expression of phase-specific genes[39,43]. As expected, and later confirmed by expression of individual cell cycle genes (Fig. 4b), HSCs were more quiescent, with almost one third of the PreME/PreMeg cells still in the G0/G1 phases (Fig. 4a). We also confirmed cluster separation between cells in G0/G1 and S/G2/M phases by performing t-SNE analysis based on all 15 measured cell cycle genes but restricted to PreME/PreMeg populations (Fig. 4c). To determine whether the expression of MEP/platelet genes is dependent on progression through the S/G2/M phases, we again used t-SNE analysis to compare PreME/PreMeg cells in the G0/G1 and S/G2/M phases. There was no separation of cells according to their cell cycle status (Fig. 4c), suggesting that PreME/PreMeg cells had

previously upregulated differentiation genes in the G0/G1 phases of the cell cycle. That PreME and PreMeg cells increase the expression of lineage-specific genes independent of cell cycle phase was further supported by comparing the mean expression of MEP/platelet genes between cells in G0/G1 and S/G2/M phases (Fig. 4d). Indeed, PreME and PreMeg cells increase the expression of the lineage-specific genes independent of cell cycle phases. These data imply that transplanted HSCs are able to differentiate before entering the S phase of the cell cycle.

To corroborate these findings, we used Ki67[RFP] knock-in mice[44]. KI67 is a nuclear protein that is absent in the G0 phase, starts to be synthesized at the beginning of the S phase, increases until mitosis, and gradually decreases thereafter in the G1 phase of the daughter cells until re-entry into the S phase[45]. We first confirmed that none of the RFP[−] cells (LSK or MP) was in the S/G2/M phase, (Supplementary Fig. 6a) and that only RFP[+] cells incorporated bromodeoxyuridine (BrdU; Supplementary Fig. 6b). Using an antibody against KI67, we found that RFP[+] expression truly reflects KI67 expression at the protein level (Supplementary Fig. 6c). Thus Ki67[RFP] knock-in mice are an appropriate tool to trace cell cycle progression in hematopoietic cells.

To follow HSCs through cell cycle progression and differentiation, we sorted RFP[−] HSCs residing in the G0/G1 phases, labeled them with CellTrace Violet, and transplanted these cells into non-conditioned recipients. Our results reveal that the majority of donor undivided MPs did not upregulate RFP expression (Fig. 4e), thus remaining in the G0/G1 phases. When taken together with the above results, these findings demonstrate that phenotypic HSCs do not require S-phase entry to become phenotypic MPs.

**Functional differences between undivided HSCs and progenitors**. We used in vitro colony assays to verify functional differences between undivided phenotypic HSCs and MPs due to changes in gene expression profiles. Undivided donor HSCs (LSK CD48[−] CD150[+]) and PreMegs (Lin[−] Sca-1[−] Kit[+] CD150[+] CD41[+]) were isolated at 36 h after transplantation and cultured as single cells in the presence of growth factors (stem cell factor (SCF), thrombopoietin, interleukin-3, and erythropoietin)[46]. Twelve days later, 89% of HSCs were multipotent and gave rise to all cell types (myeloid, erythroid, and megakaryocyte), whereas 92% of the PreMegs differentiated into megakaryocytes alone, clearly suggesting that this population had lost their multipotency (Fig. 5a).

We further investigated the in vivo repopulating capacity of donor cells. For this, we sorted undivided donor GFP[+] LSK and MP cells obtained at 36 h after transplantation of LSK CD48/41[−] CD150[+] cells, and re-transplanted the same amount of cells into

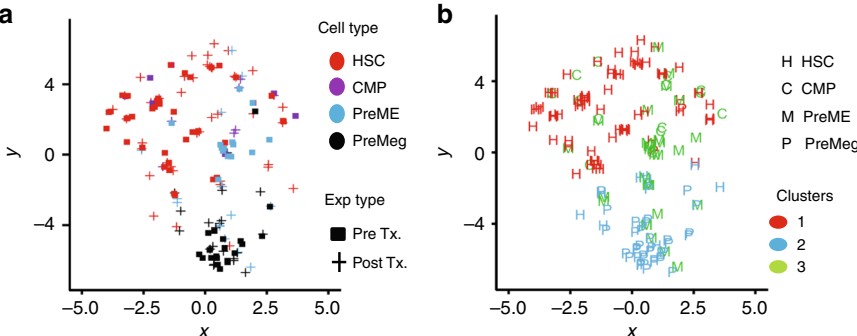

**Fig. 3** Comparison of gene expression between cells before transplantation and undivided cells after transplantation. **a** t-SNE plot for MEP/platelet genes for all cells before transplantation and undivided donor cells at 36 h after transplantation. Axes display arbitrary units. **b** t-SNE visualization for all cells before transplantation and all undivided cells after transplantation (36 h). The color coding depicts the results of a reproducible $k$-means clustering ($k = 2$) on all cells before and after transplantation based on MEP/platelet genes

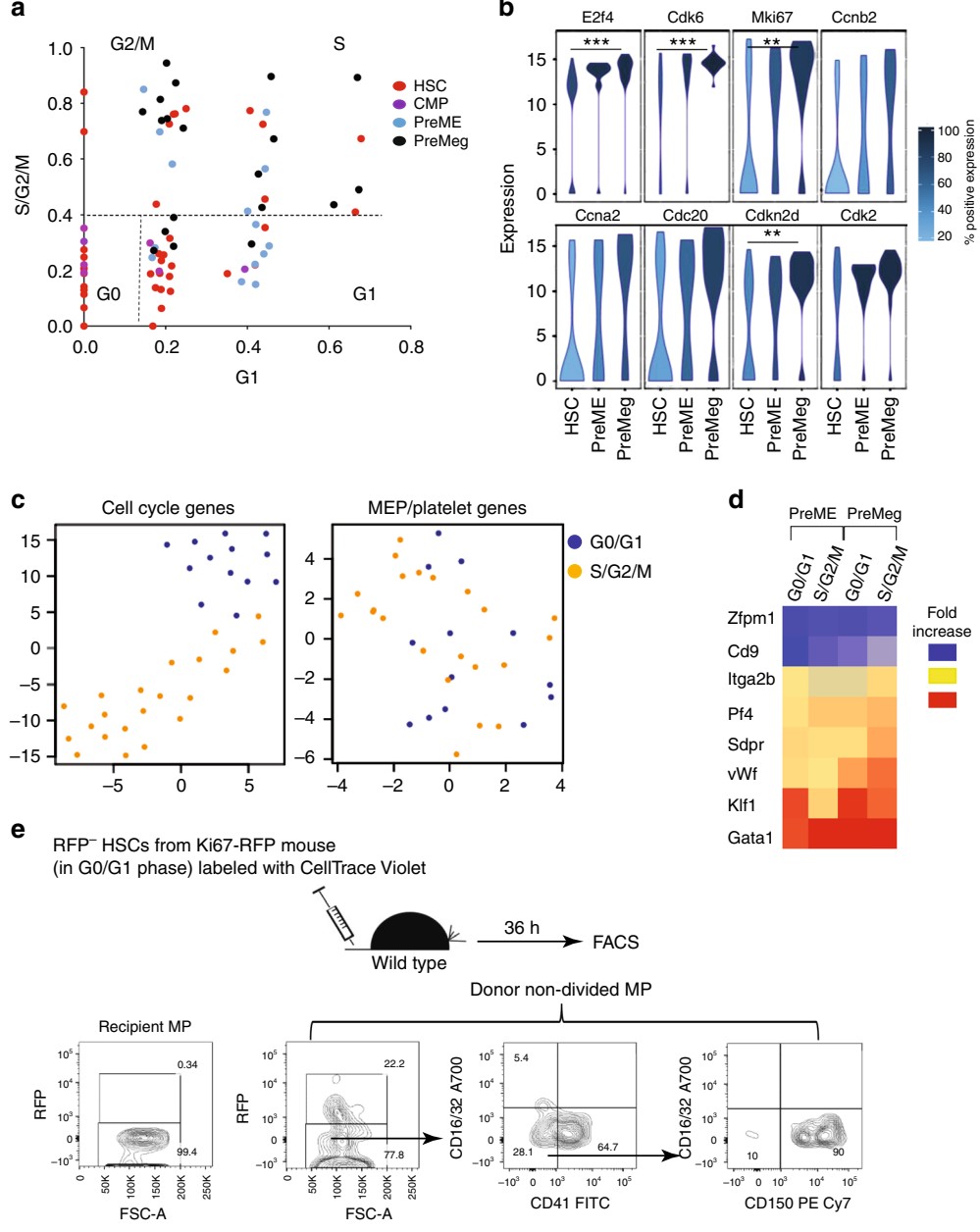

**Fig. 4** Cell cycle distribution of undivided donor HSCs, CMPs, PreMEs, and PreMegs. **a** Prediction of cell cycle phases for all undivided donor cells 36 h after transplantation. Shown is the average expression of G1 genes (x axis) and S/G2/M genes (y axes). **b** Violin density plots for the most differently expressed cell cycle genes. y Axis represents gene expression. The horizontal width of the plot shows the density of the data along the y axis. Statistical significance was determined using the Hurdle model. *P < 0.05, **P < 0.01, ***P < 0.0001. Exact P-value in supplemental Tables 2-3. **c** t-SNE plots for PreME/PreMeg cells based on cell cycle genes and MEP/platelet genes. **d** Mean expression of MEP/platelet genes was calculated for HSCs, PreMEs, and PreMegs in G0/G1 and S/G2/M phases and is depicted as fold increase relative to mean expression in HSCs in the G0/G1 phases. **e** RFP expression in undivided donor MPs at 36 h after transplantation of RFP⁻ HSCs from Ki67^RFP knock-in mice. Recipient MPs were used as negative controls for RFP expression. (representative example, n = 5, from 2 independent experiments)

lethally irradiated wild-type mice (Fig. 5b, c). Although both populations gave rise to long-lived erythroid cells, only mice transplanted with LSKs displayed donor-derived GFP⁺ short-lived neutrophils and platelets at 3 weeks after transplantation (Fig. 5b–d); moreover, LSK but not MP showed donor-derived neutrophils, platelets, erythrocytes, and lymphocytes 16 weeks after transplantation (Supplementary Fig. 6d). These observations imply that hematopoietic progenitor cells that downregulate Sca-1 without prior cell division, as expected, exhibit a dramatic reduction in their repopulation capacity.

## Discussion

In this study, we demonstrated in vivo that HSCs can differentiate into ST-HSCs, MPPs, and even restricted MPs before undergoing cell division. Using a cell-tracing approach and Ki67^RFP knock-in mice, we followed HSC differentiation in vivo and analyzed the expression of several essential megakaryocyte-erythroid- and myeloid-specific genes, and cell cycle genes, at the single-cell level. Our findings using undivided PreMegs reveal that phenotypic and gene expression changes in undivided but differentiated progenitors are accompanied by loss of multipotency and repopulation capacity after transplantation. Based on restricted

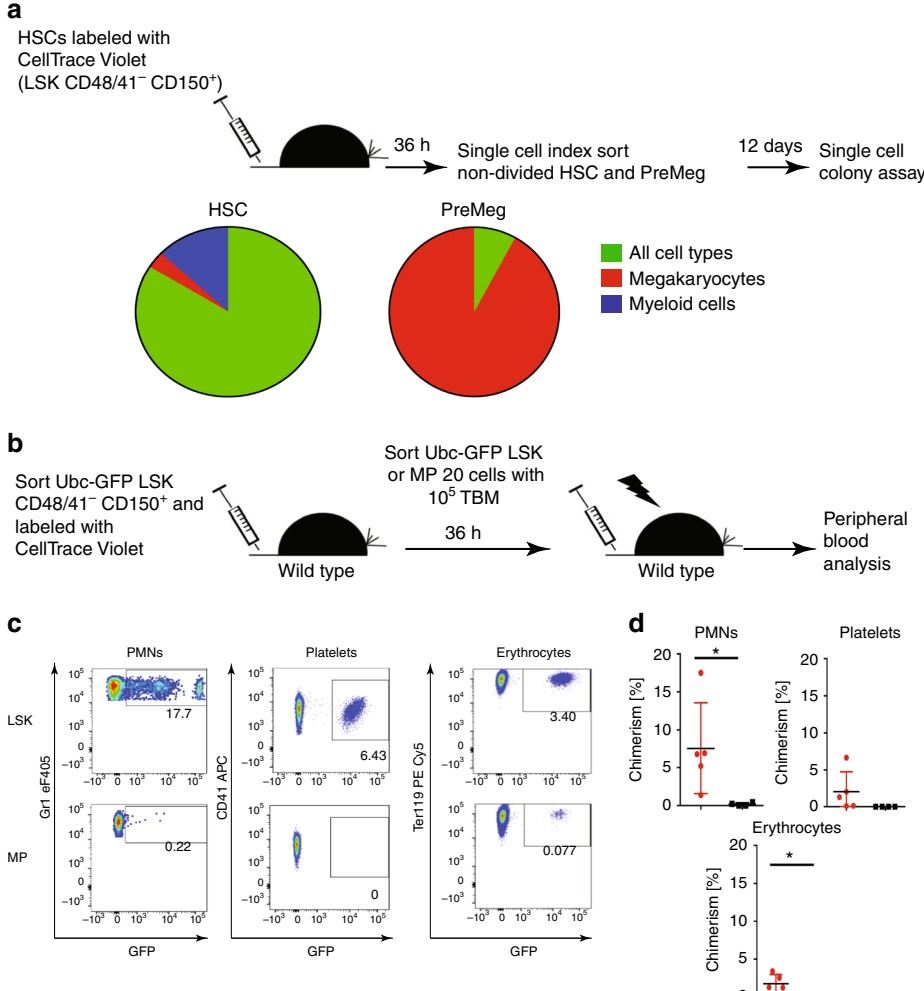

**Fig. 5** Functional analysis of undivided donor HSCs and MPs. **a** Individual undivided donor HSCs (LSK CD48⁻ CD150⁺) and PreMeg (Lin⁻ Sca-1⁻ Kit⁺ CD41⁺ CD150⁺ CD16/32⁻) cells were sorted 36 h after transplantation and cultivated in liquid culture media supplemented with mSCF, mTPO, mIl3, and hEpo. Cell composition was analyzed after 12 days using May-Grunwald–Giemsa staining. Colonies ($n = 31$) for HSCs and ($n = 25$) PreMegs, 3 independent experiments, 15 mice. In all, 82% HSCs generated colonies (>20 cells) and 79% PreMegs generated >3 megakaryocytes. **b** Reconstitution experiment using Ubc-GFP mice. **c** Peripheral blood analysis at 3 weeks after secondary transplantation into lethally irradiated recipients. Donor cell contribution to peripheral blood neutrophils (PMNs) CD11b⁺ Gr1⁺, platelets Ter119⁻CD41⁺, and erythrocytes Ter119⁺. Plots and pictures from 2 independent experiments ($n = 5$). LSK repopulation is shown in the upper panel and MP in the bottom panel. We checked the mice every 3–4 weeks for a period of 16 weeks after transplantation but did not find any repopulation from MPs. **d** Quantification of peripheral blood analysis 3 weeks after transplantation from 2 independent experiments, $n = 5$. LSK is in red, MP is in black. Statistical significance was determined using unpaired Student's $t$-test (*$P < 0.05$). Data are means $+/-$ s.d.

PreME and PreMeg progenitors as an example of differentiated cells, we reveal that HSCs can initiate a specific differentiation program in the G0/G1 phases, which is before the actual physical division of the cell.

HSCs are rare cells that give rise to numerous blood cell types through a series of intermediate progenitors[4]. Multipotent and restricted progenitors intensively proliferate, making them the key amplifiers of cell numbers in the hematopoietic system[3]. The currently accepted model of hematopoiesis holds that HSCs have to divide in order to produce multipotent and lineage-restricted progenitor populations[3,47,48]. Thus, with respect to HSCs, proliferation and differentiation are currently characterized as simultaneous processes; however, to date, no direct in vivo proof of this concept has been provided. On the contrary, it is also conceivable that proliferation and differentiation exist as two independent processes. A few in vitro studies have supported this argument and have suggested that HSC division and differentiation are parallel processes. Indeed, while Mossadegh-Keller and colleagues[49] have shown that the myeloid transcription factor PU.1 is induced during the first cell cycle after in vitro stimulation of HSCs with macrophage colony-stimulating factor, Yamamoto and colleagues[6] reported that HSCs can divide asymmetrically and give rise to restricted long-term repopulating megakaryocyte progenitors even after the first division. Kent and colleagues[50] have shown that HSCs downregulated a number of transcription factors responsible for self-renewal division and lost long-term repopulation capacity before first division in vitro. Using a single-cell sequencing approach, Yang and colleagues demonstrated that HSCs can express megakaryocyte and granulocyte-specific genes during the G1 phase of the cell cycle[51]. However, no in vivo studies on the possible uncoupling of HSC fate decision and cell cycle progression are currently available.

Indeed, the idea that cells can make fate decisions in the G1 phase of the cell cycle is not new. Pluripotent stem cells (PSCs) initiate differentiation during progression through the G1 phase[14] due to the presence of a 'window of opportunity', which is dependent on epigenetic changes that occur during that phase. On the other hand, PSCs maintain their pluripotent state during the S and G2 phases of the cell cycle, which is regulated by the cell cycle machinery but is independent of the G1 phase[17]. G1-phase-specific cell cycle regulators such as cyclin D directly regulate the localization of differentiation transcriptional factors in PSCs[52]. Our results reveal a new avenue by which the HSC fate decision process is connected with cell cycle progression in vivo. Moreover, our data are also in line with another report, which demonstrated that division and differentiation of B cells into plasma cells were temporally separated with no significant influence on each other[53].

In summary, we show that HSC division and their differentiation are probably independent processes and that HSCs make fate decisions before entering the S phase of the cell cycle.

Additionally, these results open new directions in determining similar capacities in human HSCs, as well as identifying the factors that influence these fate decisions in connection with cell cycle progression, during normal hematopoiesis and even pathologies associated with abnormal differentiation.

## Methods

**Mice**. C57BL/6 (B6), B6.SJL-PtprcaPep3b/BoyJ (B6.SJL), and Ubc:GFP mice were purchased from the Jackson Laboratory. Ki67$^{RFP}$ knock-in mice have been recently described in detail[44]. Mice (male and female) were used at an age of 8–12 weeks. HSC-CreERT/R-DTA mice were generated by crossing R26$^{DTA}$ (Gt(ROSA) 26Sor$^{tm1(DTA)Lky}$) and HSC-CreERT (Tg(Tal1-cre/ERT)42–056Jrg)[37] and used as recipients (male and female) at the age of 14–16 weeks. All mice were bred and maintained under specific pathogen–free conditions in the animal facility at the Medical Theoretical Center of the University of Technology, Dresden. Experiments were performed in accordance with the German animal welfare legislation and were approved by the "Landesdirektion Sachsen – Referat 24.1".

**Cre-activation**. One week before the start of TAM administration, mice were kept on low phytoestrogen standard diet (LASvendi, Solingen, Germany). TAM tablets 30 mg (Ratiopharm, Ulm, Germany) were dissolved overnight in lipid emulsion (SMOFlipid, Fresenius Kabi, Bad Homburg, Germany). TAM 20 mg/ml solution was applied two times (72 h apart) by oral gavage at a dose of 0.2 mg/g body weight to animals at the age of 8–14 weeks.

**Transplantation**. Bone marrow (BM) was isolated from mouse tibia, femora, pelvis, and vertebrae; crushed; and filtered through a 70-μm cell strainer. Cells were lysed in ACK Lysis Buffer (Life Technologies Cat. A10492-01) and lineage depleted using biotinylated antibodies (anti-mouse CD3 (2C11; 17A2) (1:1000/Cat. 13-0031-82), CD11b (M1/70) (1:500/Cat. 13-0112-81), CD19 (1D3) (1:500/Cat. 13-0193-81), CD45R (RA3-6B2) (1:400/Cat. 13-0452-82), Gr-1 (RB6-8C5) (1:800/Cat. 13-5931-82), Nk1.1 (PK136) (1:2000/Cat. 13-5941-81), Ter119 (1:200/Cat. MA5-17819), and anti-biotin micro-beads using magnetic cell separation (Miltenyi Biotec Germany Cat. 130-090-485). Cells were then stained with antibodies and CellTrace Violet dye (Molecular Probes Cat. C34557) according to the manu-facturer's instructions. Cells were sorted on a fluorescence-activated cell sorter (FACS) Aria II or III (BD Bioscience). In all, 3600 HSCs (Lin$^-$ Sca-1$^+$ Kit$^+$ (LSK) CD48$^-$CD41$^-$ CD150$^+$), 5000 MPP2 (LSK CD48$^+$ CD150$^+$), or 10,000 MPP3/4 (LSK CD48$^+$ CD150$^-$) cells were transplanted via intravenous injection into non-conditioned C57BL/6 mice. CD4$^+$ CD62L$^+$ naïve T cells (10$^6$), labeled with CellTrace Violet, were transplanted as controls for undivided cells. Lymph node donor cells were analyzed 36 h after transplantation along with LSK cells. For transplantation of cells from Ki67$^{RFP}$ knock-in mice, RFP$^-$ cells were sorted and donor BM cells were analyzed 36 h after transplantation, based on CellTrace Violet staining. For competitive transplantation, 20 GFP$^+$ LSK cells or MPs (Lin$^-$ Sca-1$^-$ Kit$^+$) were sorted 36 h after a primary transplantation of 3600 HSCs from Ubc-GFP mice into unconditioned recipient C57BL/6 mice. LSKs and MPs were transplanted together with 10$^5$ non-fractionated BM cells from B6.SJL mice into lethally irradiated (900 cGy) C57BL/6 wild-type recipients.

**Flow cytometry**. All analyses were done on FACS Aria II and Canto (BD Bioscience). The antibodies used for staining are mKi67 (1:100/Cat. 11-5698$^-$82), CD117 (2B8) (1:600/Cat. 47-1171$^-$80), Sca-1 (D7) (1:100/Cat. 15-5981-81), Ter119 (Ter119) (1:200/Cat. 15-5921-81), CD41 (MWReg30) (1:400 (fluorescein

isothiocyanate (FITC))–1:800 (allophycocyanin)/Cat. 11-0411-82), CD105 (MJ7/18) (1:200/Cat. 12-1051-82), CD16/32 (93) (1:50/Cat. 56-0161), CD11b (M1/70) (1:1200/Cat. 12-0112-81), Gr-1 (RB6-8C5) (1:800/Cat. 48-5931), CD3e (1:200/Cat. 17-0031-82), and CD45R (1:400/Cat. 13-0452-82) all from eBioscience. CD48 (HM48-1) (1:300/Cat. 103411) and CD150 (TC15-12F1) (1:50/Cat. 115914) are from BioLegend.

**Single-cell index sorting**. Isolated cells were single-cell sorted into 8-well strips containing 5 μl of phosphate-buffered saline. To record marker levels of each cell, the FACS Diva-7 "index sorting" function was activated during cell sorting. Using index sorting, single cells were sorted from the entire Lin$^-$ Kit$^+$ CellTrace Violet$^+$ space, and the intensities of the CellTrace Violet, Kit, Sca-1, CD41, CD48, CD150, CD105, and CD16/32 FACS markers were recorded and linked to each cell's position.

**Cytospins**. Cells were spun onto object slides at $200 \times g$, dried, and stained with May-Grunwald and Giemsa solution (Sigma Aldrich).

**In vitro culture**. Single cells were sorted and cultured in 96-well plates in StemSpan SFEM medium (STEMCELL Technologies, Cat. 09600) supplemented with 20 ng/ml rmSCF (Peprotech, 250-03), 20 ng/ml rmTPO (eBioscience, 34-8686-63), 20 ng/ml rmIl3 (Peprotech, 213-13), and 5 U/ml rhEpo (Roche) and cultivated for 12 days at 37 °C with 5% $CO_2$.

**Cell cycle analyses**. For intracellular staining, cells were fixed and permeabilized using fixation and permeabilization buffers from eBioscience. To distinguish between the G0 and G1 phase, cells were stained with intracellular Ki67 FITC (eBioscience, clone SolA15). DAPI (4, 6 diamidino-2-phenylindole; Molecular Probes) was used to measure DNA content and separate the cells in S/G2/M phases from those in the G0 and G1 phase. For the BrdU incorporation assay, 10 μM BrdU (Sigma-Aldrich) was added to the culture for 3.5 h and BrdU incorporation ana-lyses were performed using anti-BrdU-FITC ab (eBioscience, clone BU20a, Cat. # 11-5071-42)[46].

**Clustering-based analysis of cell cycle state**. Cell cycle genes were classified based on single-cell deep sequencing data[39] or defined previously in synchronized HeLa cells[43] (G1 phase genes: *Ccne1, Cdk2, Cdkn1a, Cdkn1c*; S/G2/M phase genes: *Cdkn2d, E2f4, Cdk6, Cdkn2c, Ccng2, Ccnf, Mki67, Ccna2, Ccnb1, Ccnb2, Cdc20*). First, expression of each gene for each cell was normalized to the maximum expression of the gene; second, cell cycle signature for each cell was defined as the average expression of phase-specific subsets of cell cycle genes. Discrimination between G1 and S/G2/M was done based on the distribution of control HSCs (before transplantation) and data that around 90% of HSCs (mouse strain C57Bl6) are in G0/G1 phase of the cell cycle.

**Single-cell qPCR**. Gene expression profiles of single cells were obtained using a modified protocol[54,55]. Briefly, cDNA was synthesized directly on the cells using the Quanta qScript$^{TM}$ cDNA Supermix. Total cDNA was pre-amplified for 20 cycles (1 × 95 °C 5′, 95 °C 45″, 60 °C, 1′, 72 °C 1.5′) and once at 68 °C for 10′ using the Multiplex PCR Kit (Qiagen, Hilden, Germany) in a final volume of 35 μl in the presence of primer pairs (25 nM for each primer) for all genes (listed in Table S1). Pre-amplified cDNA (10 μl) was then treated with 1.2 U Exonuclease I, and gene expression was quantified by real-time PCR on the BioMark$^{TM}$ HD System (© Fluidigm Corporation, CA, USA) using the 96.96 Dynamic Array IFC, the GE 96 × 96 Fast PCR+Melt protocol, the SsoFast EvaGreen Supermix with Low ROX (BIO RAD, CA, USA), and 5 μM primers, for each assay. Raw data were analyzed using the Fluidigm Real-Time PCR analysis software.

**Bioinformatics analysis**. Pre-processing and data analysis of single-cell expression profiles were conducted using the KNIME 2.11.2, R Version 3.3.2, and RStudio Version 0.99.486 and version 1.0.136 (Boston, MA, USA) software. Where further required, pre-processing via a linear model to correct for confounding sampling effects was conducted[54]. t-SNE plots were created using the R package "Rtsne". To model the bi-modal gene expression of single cells, the Hurdle model, a semi-continuous modeling framework, was applied to pre-processed data[56]. This allowed us to assess differential expression profiles as a function of frequency of expression and mean positive expression using a likelihood ratio test. $k$-means clustering for $k = 2$ was performed on the normalized data and using the R package "stats".

**Statistical analysis**. Data were expressed as mean $+/-$ standard deviation (s.d.). Statistical analyses based on unpaired Student's $t$-test were performed using the Prism 5.0 software (GraphPad). $P$-value <0.05 were considered as statistically significant.

**Data availability**. All data generated or analyzed during this study are included in this published article and its Supplementary Information files or are available from the corresponding authors upon reasonable request.

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

## Acknowledgements

T.G. received support from the Fritz Thyssen foundation (10.14.2.153). B.W. was supported by the Heisenberg program (Deutsche Forschungsgemeinschaft – DFG, Germany; WI3291/ 5-1). This work was supported by grants from the DFG (SFB655 "Cells into Tissues" to T.C. and GR4857/1-1 to T.G.), a CRTD seed grant to T.G. and B.W., and a European Research Council grant (683145) to T.C. The work of L.T. and I.G. was supported by the German Federal Ministry of Research and Education, Grant number 031A315 "*MessAge*". We would like to thank Dr. Vasuprada Iyengar for critically reading the manuscript.

## Author Contributions

Conceptualization: T.G. and B.W.; investigation: A.K., S.D., and T.G.; methodology: T.G., A.E., L.T., B.R., M.v.B., A.G., and I.G.; resources: T.G., O.B., H.C., and B.W.; writing original draft and creating figures: T.G. and B.W.; writing, review, and editing: T.G., A.E, I.G., L.T., O.B., T.C., and B.W.; funding acquisition: T.G., L.T., I.G., T.C., and B.W.; supervision: T.G., T.C., and B.W.

## Additional information

**Competing interests:** The authors declare no competing interests.

