## [Peer Review File · Nature Communications]

Reviewers' comments:

Reviewer #1 (Remarks to the Author):

Grinenko et al use a non-irradiation haematopoietic stem cell (HSC) mouse transplantation model to argue that HSCs can differentiate into progenitor cells before undergoing cell division. The authors find lineage markers are upregulated on a percentage of non-divided HSC and that these changes are associated with gene expression changes, notably an upregulation of genes associated with differentiation. Major reservations about this study are does this process occur in native (non-transplantation) conditions, and what is the physiological relevance of HSC differentiation without cell division? While these may be difficult to answer, a related and more tractable question is does this process occur in normal HSC transplantation assays using irradiated recipients? If HSCs also lose multipotency via this mechanism within this assay, these findings could have important implications for how the field interprets results from such experiments. Also, it is important to understand whether the HSC differentiation a result of transplanting into a recipient with a "full" HSC niche or due to ex vivo isolation. This could be directly tested by transplanting HSCs into mice with depleted HSCs (e.g. Kit mutant/KO mice), and would significantly strengthen this manuscript.

Specific comments:

1. Title: should be updated to more accurately reflect the data presented.
2. Main text: the complex experiments performed are often difficult to follow and the authors should also be careful with terminology as it is often not clear exactly what cell population be being described. These should be improved.
3. The route of HSC transplantation is not detailed in the paper. Where HSCs were transplanted by intra-femoral (IF) injection or via tail vein (or other) alters the in vivo environment that HSCs find themselves. This could have important implications on extracellular signaling, and therefore cell fate. If not undertaken, IF should be tested.
4. Line 29 - Last sentence of the summary should be more accurately written.
5. Haas et al Cell Stem Cell and Yamamoto et al Cell deserve to be discussed in the introduction.
6. Line 59 – remove inappropriate comma.
7. Figure 1A-C. It would be useful to display the FACS purified donor HSC population transplanted – as a side-by-side comparison to the immunophenotype after 36 hours.
8. Figure 1B – axis title missing.

9. Wilson et al Cell Stem Cell should be referenced when discussing the correlation between Sca1 expression and HSC function (line 65).
10. Figure 1C. First, can axis be altered to improve resolution of the two populations? Second, given the bi-modal overlapped distribution of the populations, it might be more accurate to compare CellTrace-high and CellTrace-low populations, rather than simply dividing the population in two.
11. Line 79-81: These are phenotypic HSCs and phenotypic progenitors.
12. Line 90: It should be made clear the cells used in the transplant experiment.
13. Separation of cell types based on gene expression is clear for freshly isolated cells types, but less clear for 36h transplanted cells. What happens if you use the fresh cell plots as “reference plots” and overlay the transplanted qPCR data onto them (such as done by Scialdone et al Nature)? Do the phenotypic cell types correspond?
14. Figure 3B. It is not clear how these clusters were generated. Can this be linked to a supplementary figure?
15. Line 116: correct sentence.
16. Line 122: use of “upregulated” is misleading as the authors don’t know what point during differentiation/cell cycle, this upregulation occurred.
17. Figure 4A – what percentage of cells generated colonies? Is this different between the HSC and MP populations?
18. Figure 4C – what is the long-term contribution of these cell populations in vivo? As only 20 primitive stem/progenitor cells were transplanted, in vivo output may not be present at such an early time point (3 weeks).
19. Figure 4C – why was MP compared to LSK and not phenotypic HSCs?
20. Figure 4C – HSC function is defined by multipotency into both myeloid and lymphoid lineages. What is the lymphoid potential of these cells?
21. Figure 4 – how do the authors explain the MK bias in vitro but erythroid bias in vivo?
22. Line 190 – “cell division” might be more accurate than “proliferation” as the authors only really focus on the first cell division.
23. Discussion: How does this work fit with recent findings by Bernitz et al Cell 2016?
24. Methods: What buffer was used during the HSC isolation? Use of FBS supplemented PBS, for example, could result in HSC stimulation/commitment. If so, does the same upregulation of differentiation markers occur in vitro before cell division? I.e. does the HSC differentiation initiate ex vivo?

Reviewer #2 (Remarks to the Author):

This paper seeks to address a longstanding question of interest to the field of cell biology, which seeks to understand how lineage potential is set up and commitment established, and the role of cell division in these mechanisms. HSC biology is an attractive system to address these issues because of the potential clinical relevance of the findings and the accessibility of the cells and systems to analyze their differentiation. At the same time, this system is also tricky to obtain definitive data from, because HSC are rare and heterogeneous both biologically and molecularly at the epigenetic, RNA and protein levels. In addition, it is no longer clear what in vivo read-outs post-transplant are detecting in terms of lineage decision processes by responsive cells. In this paper, the authors introduce a number of novel aspects to their experimental design that lend interest to the findings (use of unirradiated recipients, Ki67RFP donors). Nevertheless, interpretation of the results is significantly weakened by several critical untested assumptions, as summarized below.

1. It is assumed that the label-retaining cells are undivided based on label retention and gene expression. But the label level in many divided cells overlaps with that seen in many undivided cells. Thus it is important to provide more convincing quantitative data to exclude the possibility that the observations reflect those of rare cells that initially incorporated a higher level of label and did divide, given that division only results in a 2-fold dilution of label. Alternative strategies to demonstrate lack of division (eg insensitivity to cell-cycle-specific drugs, Fucci labeling) would also help to make the case made using phenotype data to detect differentiation more convincing.
2. It is also assumed that the frequency of cells that had multipotent functional HSC potential within the phenotype transplanted into primary mice was sufficiently high to be sure that these were the cells that “differentiated” directly. I did not see any data measuring this initial critical value to compare with the frequency of directly differentiated cells detected. Indeed, I did not see any data to validate the purity of the HSC phenotypes isolated. Given that each mouse received 3600 cells of the HSC phenotype, even a small level of contamination could give the result seen.
3. In conradistinction to the claims made in paragraph 2 of the Discussion, definitive evidence of “differentiation” of HSC before division has been previously documented, see Kent et al Blood 112:560-7, 2008. This paragraph needs rewriting and this reference should be added.
4. The conclusion that the present study shows that lineage commitment decisions in pluripotent embryonic cells and HSCs use similar mechanisms as stated in the Abstract and Discussion is inappropriate as no mechanistic evidence for this statement is provided here. These statements are mere speculations.

5. Please also discuss how these findings relate to those recently reported by Shimoto et al in Blood 2017 129:2124-31.

6. Please provide details of the numbers of cells analyzed to generate the data in each figure, including those that failed to yield useful data.

Minor

1. line 34 – this statement needs qualification. It is well documented that some HSCs do divide even in adults and that most if not all are dividing in the fetus.

Dear Editor,

We would like to thank you for your decision letter from 29th of August. Moreover, we would like to thank you and the reviewers for the careful scrutiny of our manuscript and their helpful comments and suggestions for improvement. We have thoroughly addressed all the different issues raised by the reviewers and have revised our manuscript accordingly. This required substantial additional experimentation, as evidenced by several new results and figure panels added to the manuscript. We provide here a point-by-point response to the questions of both reviewers.

We firmly believe that our revised manuscript establishes that HSCs are able to differentiate without undergoing cell division, therefore uncoupling fate decisions from cell division. We thank you in advance for considering our revised manuscript and hope that we can convince you and the reviewers of the impact of our findings and that our work will be acceptable for publication in *Nature Communications* in order to reach its broad readership.

Reviewer #1 (Remarks to the Author):

Grinenko et al use a non-irradiation haematopoietic stem cell (HSC) mouse transplantation model to argue that HSCs can differentiate into progenitor cells before undergoing cell division. The authors find lineage markers are upregulated on a percentage of non-divided HSC and that these changes are associated with gene expression changes, notably an upregulation of genes associated with differentiation. Major reservations about this study are does this process occur in native (non-transplantation) conditions, and what is the physiological relevance of HSC differentiation without cell division? While these may be difficult to answer, a related and more tractable question is does this process occur in normal HSC transplantation assays using irradiated recipients? If HSCs also lose multipotency via this mechanism within this assay, these findings could have important implications for how the field interprets results from such experiments. Also, it is important to understand whether

the HSC differentiation a result of transplanting into a recipient with a “full” HSC niche or due to ex vivo isolation. This could be directly tested by transplanting HSCs into mice with depleted HSCs (e.g. Kit mutant/KO mice), and would significantly strengthen this manuscript.

We thank the reviewer for bringing up this very important issue, as this indeed helps highlight the impact of our presented approach. In our study, we used unconditioned recipients for transplantation of the HSCs to avoid any additional stress (e.g. irradiation). Irradiation of recipient mice leads to the generation of free radicals, including reactive oxygen and nitrogen species, a cytokine storm as well as severe temporary but also permanent damage to the bone marrow niche cells ^{1,2}. Expectedly, these changes will dramatically influence the behavior of transplanted HSCs. The question how HSCs will behave in "normal transplantation assays", using irradiated recipients is extremely interesting but at the same time very challenging. Nevertheless, we have now tested this setting using recipients receiving a lethal (9 Gy) or sub-lethal dose of radiation (1Gy). To discriminate between HSCs/multipotent progenitors (MPP) and myeloid progenitors, we utilized Kit and Sca-1 expression, as these are the only useful surface markers currently available. However, and as shown in Response Figure 1, the expression of both markers was significantly disturbed on the donor HSCs, 36 hours after transplantation (Response Figure 1). Therefore, we have to conclude that it is technically impossible to analyze early differentiation events, as described in our manuscript for unconditioned recipients, when transplantation of HSCs is performed into irradiated recipients.

Response Figure 1. Transplantation into irradiated recipients. HSCs (LSK CD48⁻ CD41⁻ CD150⁺) or LSK cells from UBC-GFP strain were transplanted into irradiated recipients. 36h after transplantation donor GFP⁺ cells were analyzed. **(a)** Non-irradiated control for surface staining. **(b)** Donor cells after transplantation 6x10⁴ LSK cells. **(c)** Donor cells after transplantation 4000 HSCs.

The reviewer's point on the "full" or "empty" HSC niche in the bone marrow represents one of the major questions in the field that is currently under debate^{3,4}. Previous studies suggested that occupied niche space by endogenous HSCs (i.e. a "filled" niche) prevented successful engraftment of transplanted HSCs without myelosuppressive host conditioning⁵. Recently, it was shown that upon transplantation of a large amount of HSCs in non-conditioned recipients, donor HSCs occupied empty niches that were distant from niches in which endogenous host HSCs resided. These donor HSCs could generate hematopoietic progenitors and peripheral blood cells in a normal fashion⁴. In our recent work, we have also shown that massive reduction of HSCs, using a specific model of HSC depletion, does not induce expansion of residual HSCs despite the existing empty niche space⁶. To answer the reviewer's question and to analyze whether a low number of endogenous HSCs can potentially influence the differentiation of transplanted HSCs, we used the HSC-CreERT-R26^{DTA/DTA} mouse line⁶ (New supplementary Figure 1f). In these mice, HSCs were depleted by oral administration of tamoxifen (TAM). Three weeks after induction of Cre-recombinase, mice had four-fold fewer HSCs compared with Cre-negative control mice (New supplementary Figure 1f). Next, HSCs from wild-type mice were sorted, labeled with CellTrace dye, and transplanted into the aforementioned mice, as described in our manuscript. Interestingly, the frequency of donor-derived myeloid progenitors was independent of the amount of endogenous HSCs (New supplementary Figure 1g). We also did not find any difference in the frequency of generated restricted progenitors between Cre- and Cre+ mice (New supplementary Figure 1h). Furthermore, donor HSCs generated more undivided GMPs in all TAM-treated mice compared with non-TAM treated mice, a phenomenon that is most probably induced by the additional tamoxifen-stress. In conclusion, these additional experiments are very informative as they reveal that only non-irradiated recipients are suitable to study early differentiation events, whereas they are independent of the recipient's HSCs.

Specific comments:

1. Title: should be updated to more accurately reflect the data presented.

We thank the reviewer for this suggestion and have now changed the title to: “Hematopoietic stem cells differentiate into restricted myeloid progenitors before cell division”.

2. Main text: the complex experiments performed are often difficult to follow and the authors should also be careful with terminology as it is often not clear exactly what cell population be being described. These should be improved.

We apologize for any the confusion that we might have introduced; in the revised manuscript, we have clearly defined the phenotype of all analyzed populations in the main text or figure legends.

3. The route of HSC transplantation is not detailed in the paper. Where HSCs were transplanted by intra-femoral (IF) injection or via tail vein (or other) alters the in vivo environment that HSCs find themselves. This could have important implications on extracellular signaling, and therefore cell fate. If not undertaken, IF should be tested.

All HSC transplantations were performed via intravenous injection (iv). We have now added this information to the Materials and Methods section. As suggested by the reviewer, we have also tested the method of intra-femoral (IF) injection. Although we now reveal that IF transplanted HSCs can indeed differentiate into myeloid progenitors without cell division, the number of donor cells was not sufficient to analyze the frequency of the different restricted myeloid progenitors (Response Figure 2). A possible explanation for this negative result lays in the fact that the bone marrow environment is seriously disturbed after the IF injection, and consequently can lead to the death of a majority of transplanted HSCs⁷. In previous studies, repopulation of IF-transplanted cells was detected only several months after transplantation into irradiated recipients^{3,8}. In such settings, even small amounts of surviving cells will expand and can be detected later. However, to the best of our knowledge there is currently no

publication available describing the analysis and purification of HSCs within a time frame of 1-2 days after IF transplantation.

The fact that our specific iv-transplantation method allows HSCs to make a fate decision before progressing into the S phase of the cell cycle, hence, before cell division, is a key finding of our work. Although we performed the IF experiments, due to technical restrictions and the poor outcome of the IF model, as explained above, we decided not to include this part into the new version of our manuscript but share the results of our additional efforts with the reviewer via this rebuttal letter (Response Figure 2). Nevertheless, we do acknowledge the importance of the question that different conditions could indeed lead to different lineage choices.

Response Figure 2. IF transplantation of HSCs. 4000 GFP+ CellTrace Violet labeled HSCs were transplanted via IF injection, and donor cells were analyzed 36h later. A representative plot of three independent experiments is shown (all cells displayed were collected from 10 different legs).

4. Line 29 - Last sentence of the summary should be more accurately written.

We re-wrote this sentence.

5. Haas et al Cell Stem Cell and Yamamoto et al Cell deserve to be discussed in the introduction.

We agree with the reviewer and discussed these publications in the introduction.

6. Line 59 – remove inappropriate comma.

We removed the comma.

7. Figure 1A-C. It would be useful to display the FACS purified donor HSC population transplanted – as a side-by-side comparison to the immunophenotype after 36 hours.

We appreciate this remark and added the FACS plots of donor HSCs to the main Figures (Figure 1A).

8. Figure 1B – axis title missing.

We have now added the axis title.

9. Wilson et al Cell Stem Cell should be referenced when discussing the correlation between Sca1 expression and HSC function (line 65).

We cited this publication related to the statement.

10. Figure 1C. First, can axis be altered to improve resolution of the two populations? Second, given the bi-modal overlapped distribution of the populations, it might be more accurate to compare CellTrace-high and CellTrace-low populations, rather than simply dividing the population in two.

We thank the reviewer for this comment and have carefully considered the different options. However, the proposed alteration of the axis does not improve the resolution of the two populations. In addition, we re-analyzed our data as suggested by the reviewer and compared CellTrace-high and CellTrace-low populations to avoid overlap of cells that had passed one cell division with non-divided cells with diluted CellTrace dye. Interestingly, using this new gating strategy, we found no difference in the frequency of restricted progenitors compared to our previous method. We have added this alternative approach to the revised version of our manuscript (New supplemental Figure 1d-e).

11. Line 79-81: These are phenotypic HSCs and phenotypic progenitors.

We agree with the reviewer and changed the sentence accordingly.

12. Line 90: It should be made clear the cells used in the transplant experiment.

We added this information to the new version of our manuscript.

13. Separation of cell types based on gene expression is clear for freshly isolated cells types, but less clear for 36h transplanted cells. What happens if you use the fresh cell plots as “reference plots” and overlay the transplanted qPCR data onto them (such as done by Scialdone et al Nature)? Do the phenotypic cell types correspond?

This is an important comparison, which we have implemented in Figure 3a. For this, we performed a t-SNE analysis of cells before and after transplantation in the same plot. Our analysis shows that cells before and after transplantation cluster very well together based on their gene expression profile and surface phenotype; these criteria were used for the classification of HSCs and restricted progenitors. Therefore, the gene expression for transplanted cells and fresh cells corresponds to their cell phenotype.

14. Figure 3B. It is not clear how these clusters were generated. Can this be linked to a supplementary figure?

The authors thank the reviewer for pointing out the partly incomplete description of the methods underlying Figure 3b. The figure is the result of a combination of two different methods, namely t-SNE and k-means clustering. t-SNE is a standard method to obtain lower dimensional (2D) representations of higher-dimensional data. The algorithm is based on a variation of Stochastic Neighbor Embedding, which aims for a reduction of dimensionality but at the same time conserving local and global structures of the respective data set. We use the method to access gene expression similarity between all analyzed cells. To further investigate the similarity between different groups of cells, we used the k-means cluster algorithm. We wondered whether HSCs and PreMegs truly

form distinctive subgroups in terms of their gene expression profile. Therefore, we excluded the intermediate cell differentiation stages (colored in green) and provided the algorithm with a number of expected clusters ($k = 2$). Figure 3b illustrates that not only the visual inspection of the t-SNE visualization but also the k-means cluster algorithm is able to distinguish between those two cell types.

In the manuscript text, we moved the reference pointing to Figure 3a-b to appropriately link it to the applied methodology. We also added a missing reference for the t-SNE approach in the methods section.

15. Line 116: correct sentence.

We corrected the sentence.

16. Line 122: use of “upregulated” is misleading as the authors don’t know what point during differentiation/cell cycle, this upregulation occurred.

We changed “upregulated” to “increased expression”.

17. Figure 4A – what percentage of cells generated colonies? Is this different between the HSC and MP populations?

82% of sorted non-divided HSCs generated colonies (containing more than 20 cells) and 79% of non-divided PreMegs generated more than 3 megakaryocytes. These results reveal that there was no substantial difference in the potency between HSCs and preMegs to generate colonies. We added this information to the figure legend of Figure 5.

18. Figure 4C – what is the long-term contribution of these cell populations in vivo? As only 20 primitive stem/progenitor cells were transplanted, in vivo output may not be present at such an early time point (3 weeks).

We checked the mice every 3-4 weeks for a period of 16 weeks after transplantation but did not find any repopulation from MPs. We added a comment to the new version of the manuscript

19. Figure 4C – why was MP compared to LSK and not phenotypic HSCs?

We appreciate the reviewer's question on this topic. After transplantation of 3600 HSCs into non-conditioned recipients, we could only analyze a very small amount of cells from each mouse (50-80 cells from all bones per mouse). Further sorting of restricted populations led to an additional loss of about 80% of cells due to our very strict gating strategy and sorting procedure. Since the main goal of this particular experiment was to estimate the repopulation capacity of the MPs, which down-regulated Sca-1 expression, we compared them with the LSK population 36h after transplantation. Importantly, this particular population contains about 50% LSK CD48⁻ CD150⁺ - HSC cells. Nevertheless, our analysis showed that even 20 of these LSK cells can better repopulate than 20 MPs.

20. Figure 4C – HSC function is defined by multipotency into both myeloid and lymphoid lineages. What is the lymphoid potential of these cells?

In the course of this project, we have also checked this population but never detected any lymphoid progenitor cells 36h after transplantation of HSCs into non-conditioned recipients. Therefore, the main goal of this study was to analyze the repopulation capacity of cells, which down-regulated Sca-1 and obtained the phenotype of myeloid progenitors 36h after transplantation of HSCs. MPs can only differentiate into erythrocytes, platelets, and myeloid cells but not into lymphoid cells. Since donor LSKs were able to give rise to erythrocytes, platelets and myeloid cells, even after two consecutive transplantations in 36h, we can exclude that loss of repopulation capacity is due to the handling of the cells.

21. Figure 4 – how do the authors explain the MK bias in vitro but erythroid bias in vivo?

We thank the reviewer for addressing this issue. For the experiment in Figure 5, we sorted only PreMeg cells, which are platelet bias. For the in vivo experiments, we transplanted MPs, cells that can still produce erythrocytes, platelets, and neutrophils. However, erythroid cells are relatively long-lived cells (40-50 days in mice) compared with platelets (5 days) or neutrophils (3h-3days). Since we analyzed the peripheral blood 3 weeks after transplantation, we could only detect long-lived erythrocyte progeny. However, this does not necessarily imply an erythroid bias of the transplanted MP cells.

22. Line 190 – “cell division” might be more accurate than “proliferation” as the authors only really focus on the first cell division.

We thank the reviewer for this suggestion and changed 'proliferation' into 'cell division' in the text.

23. Discussion: How does this work fit with recent findings by Bernitz et al Cell 2016?

Bernitz et al ⁹ have studied labeling retaining long-term HSCs (LR-HSCs); a very rare population (about 3% of LT-HSCs), which resides on top of the hematopoietic hierarchy. Based on their model, LR-HSCs perform only four self-renewal divisions and do not undergo differentiation under steady state. LR-HSCs, which divided four times represented 1.5-2% of LT-HSCs. Bernitz et al did not study how many times active HSCs can divide. In our work, we studied the differentiation capacity of HSCs but did not discriminate between LR-HSCs and active HSCs. However, about 30% of the transplanted HSCs differentiated into myeloid progenitors and about 30-40% of these without division. Therefore, our results cannot be explained by the potential differentiation of LR-HSCs that passed already four divisions and would therefore be incapable to divide anymore.

Moreover, Bernitz et al also showed that active CD41+ HSCs are enriched for myeloid-restricted progenitors, but we only used CD41- HSCs for transplantation. We added this reference to the main text of the new manuscript.

24. *Methods: What buffer was used during the HSC isolation? Use of FBS supplemented PBS, for example, could result in HSC stimulation/commitment. If so, does the same upregulation of differentiation markers occur in vitro before cell division? I.e. does the HSC differentiation initiate ex vivo?*

We carefully evaluated this important question. For the HSC isolation, we used a PBS/5% FCS buffer, and kept cells on ice during all procedures. All cells that were used as a control for gene expression analysis underwent exactly the same purification procedure (New supplementary Figure 4). After sorting and before transplantation, HSCs were reanalyzed and shown to keep the phenotype: Lin⁻ Kit⁺ Sca-1⁺ CD48⁻ CD41⁻ CD150⁺ (Figure 1a). HSCs and progenitors before and 36h after transplantation clustered together based on gene expression profile and their surface phenotype, which was used for classification of HSCs and progenitors (Figure 3). Moreover, we also analyzed sorted HSCs that we kept on ice for 36h. Importantly, we could not find any change in phenotype, strongly suggesting that only the BM environment in the mouse but not the isolation procedure has led to differentiation of the transplanted HSCs. We added these findings to Supplementary Figure 1a.

Reviewer #2 (Remarks to the Author):

This paper seeks to address a longstanding question of interest to the field of cell biology, which seeks to understand how lineage potential is set up and commitment established, and the role of cell division in these mechanisms. HSC biology is an attractive system to address these issues because of the potential clinical relevance of the findings and the accessibility of the cells and systems to analyze their differentiation. At the same time, this system is also tricky to obtain definitive data from, because HSC are rare and heterogeneous both biologically and molecularly at the epigenetic, RNA and protein levels. In addition, it is no longer clear what in vivo read-outs post-transplant are detecting in terms of lineage decision processes by responsive cells. In this paper,

the authors introduce a number of novel aspects to their experimental design that lend interest to the findings (use of unirradiated recipients, Ki67RFP donors). Nevertheless, interpretation of the results is significantly weakened by several critical untested assumptions, as summarized below.

1. It is assumed that the label-retaining cells are undivided based on label retention and gene expression. But the label level in many divided cells overlaps with that seen in many undivided cells. Thus it is important to provide more convincing quantitative data to exclude the possibility that the observations reflect those of rare cells that initially incorporated a higher level of label and did divide, given that division only results in a 2-fold dilution of label. Alternative strategies to demonstrate lack of division (eg insensitivity to cell-cycle-specific drugs, Fucci labeling) would also help to make the case made using phenotype data to detect differentiation more convincing.

We thank the reviewer for this very important question. To avoid potential overlap between divided and non-divided cells we re-analyzed our data and gated on CellTrace high from the right peak and CellTrace low from the left peak (New supplementary Figure 1d). We confirmed that HSCs give rise to restricted myeloid progenitors (CMPs, GMPs, PreMEs, and PreMegs) with the same frequency as with our initial gating strategy (Figure 1e and New supplementary Figure 1e).

Cell cycle specific drug incorporation like BrdU or EdU as well as Fucci mice would allow researchers to follow cell cycle progression but not the division rate. The dilution of BrdU or EdU does not give sharp enough peaks following single division¹⁰.

As suggested by the reviewer, we have also tested the Fucci mouse (B6.Cg-Tg(FucciS/G2/M)#492Bsi) to analyze the cell cycle progression of hematopoietic cells¹¹. However, this strain is not useful for the analysis of the cell cycle progression of HSCs due to inappropriate expression of fluorescent proteins during cell cycle transition (data not shown). In our current study, we have used the recently described Ki67-RFP knock-in strain and confirmed that this strain reflects the cell cycle progression in hematopoietic stem and progenitor cells in an appropriate way (Supplemental Figure 6). Indeed, this mouse strain has made it possible to transplant HSCs in the G0 phase and

obtain Ki67⁻ MPs, thus before entering the S/G2/M phases of the cell cycle, 36h later (Figure 4e).

2. It is also assumed that the frequency of cells that had multipotent functional HSC potential within the phenotype transplanted into primary mice was sufficiently high to be sure that these were the cells that “differentiated” directly. I did not see any data measuring this initial critical value to compare with the frequency of directly differentiated cells detected. Indeed, I did not see any data to validate the purity of the HSC phenotypes isolated. Given that each mouse received 3600 cells of the HSC phenotype, even a small level of contamination could give the result seen.

We thank the reviewer for giving us the opportunity to answer this important question. The purity of our sorted cells has always been 99%. Therefore, it is very unlikely that the 30-35% donor Lin⁻ Kit⁺ Sca-1⁻ cells (40% non-divided) 36h after transplantation are residing from a potential contamination or minor (1%) impurity in our sorted HSCs. In the revised version of our manuscript, we provide information on the purity of transplanted HSCs as well as in the legend of Figure 1a. In addition, we were unable to find any donor cell, 36h after transplantation of only Lin⁻ Kit⁺ Sca-1⁻ cells. We added this particular finding in the revised version of our manuscript (New supplementary Figure 1b).

3. In contradiction to the claims made in paragraph 2 of the Discussion, definitive evidence of “differentiation” of HSC before division has been previously documented, see Kent et al Blood 112:560-7, 2008. This paragraph needs rewriting and this reference should be added.

We have now rewritten this particular paragraph and added the requested publication to the discussion. Moreover, we added another reference demonstrating that HSCs can express megakaryocyte and granulocyte-specific genes during the G1 phase of the cell cycle¹².

4. The conclusion that the present study shows that lineage commitment decisions in pluripotent embryonic cells and HSCs use similar mechanisms as stated in the Abstract and Discussion is inappropriate as no mechanistic evidence for this statement is provided here. These statements are mere speculations.

We carefully re-formulated this statement.

5. Please also discuss how these findings relate to those recently reported by Shimoto et al in Blood 2017 129:2124-31.

Shimoto et al have shown that numerous empty HSC niches are available upon transplantation into non-conditioned recipients. These are located distant from filled niches and available for HSCs engraftment and proliferation. Donor HSCs give rise to all blood cells without any bias⁴. We discussed this statement in the results section and added the reference.

6. Please provide details of the numbers of cells analyzed to generate the data in each figure, including those that failed to yield useful data.

We have added this information to the results.

Minor

1. line 34 – this statement needs qualification. It is well documented that some HSCs do divide even in adults and that most if not all are dividing in the fetus.

We re-wrote this sentence.

Additional References:

- 1 Abbuehl, J. P., Tatarova, Z., Held, W. & Huelsken, J. Long-Term Engraftment of Primary Bone Marrow Stromal Cells Repairs Niche Damage and Improves

- Hematopoietic Stem Cell Transplantation. *Cell stem cell* **21**, 241-255 e246, doi:10.1016/j.stem.2017.07.004 (2017).
- 2 Cao, X. *et al.* Irradiation induces bone injury by damaging bone marrow microenvironment for stem cells. *Proc Natl Acad Sci U S A* **108**, 1609-1614, doi:10.1073/pnas.1015350108 (2011).
- 3 Zhong, J. F., Zhan, Y., Anderson, W. F. & Zhao, Y. Murine hematopoietic stem cell distribution and proliferation in ablated and nonablated bone marrow transplantation. *Blood* **100**, 3521-3526, doi:10.1182/blood-2002-04-1256 (2002).
- 4 Shimoto, M., Sugiyama, T. & Nagasawa, T. Numerous niches for hematopoietic stem cells remain empty during homeostasis. *Blood* **129**, 2124-2131, doi:10.1182/blood-2016-09-740563 (2017).
- 5 Bhattacharya, D. *et al.* Niche recycling through division-independent egress of hematopoietic stem cells. *J Exp Med* **206**, 2837-2850, doi:10.1084/jem.20090778 (2009).
- 6 Schoedel, K. B. *et al.* The bulk of the hematopoietic stem cell population is dispensable for murine steady-state and stress hematopoiesis. *Blood* **128**, 2285-2296, doi:10.1182/blood-2016-03-706010 (2016).
- 7 Lapidot, T., Dar, A. & Kollet, O. How do stem cells find their way home? *Blood* **106**, 1901-1910, doi:10.1182/blood-2005-04-1417 (2005).
- 8 Kent, D. G., Dykstra, B. J., Cheyne, J., Ma, E. & Eaves, C. J. Steel factor coordinately regulates the molecular signature and biologic function of hematopoietic stem cells. *Blood* **112**, 560-567, doi:10.1182/blood-2007-10-117820 (2008).
- 9 Bernitz, J. M., Kim, H. S., MacArthur, B., Sieburg, H. & Moore, K. Hematopoietic Stem Cells Count and Remember Self-Renewal Divisions. *Cell* **167**, 1296-1309 e1210, doi:10.1016/j.cell.2016.10.022 (2016).
- 10 Foudi, A. *et al.* Analysis of histone 2B-GFP retention reveals slowly cycling hematopoietic stem cells. *Nat Biotechnol* **27**, 84-90, doi:10.1038/nbt.1517 (2009).
- 11 Sakaue-Sawano, A. *et al.* Visualizing spatiotemporal dynamics of multicellular cell-cycle progression. *Cell* **132**, 487-498, doi:10.1016/j.cell.2007.12.033 (2008).
- 12 Yang, J. *et al.* Single cell transcriptomics reveals unanticipated features of early hematopoietic precursors. *Nucleic acids research* **45**, 1281-1296, doi:10.1093/nar/gkw1214 (2017).

REVIEWERS' COMMENTS:

Reviewer #1 (Remarks to the Author):

The authors have clearly gone to considerable effort to revising this manuscript, and their comprehensive responses are highly appreciated by this reviewer. The manuscript and figures are now easier to follow and understand. The paper suggests an interesting concept for the HSC field that deserves publication, although definitive evidence using heterogeneous HSC populations is challenging, and physiological relevance of this mechanism remains unclear.

The following reservations remain:

1. Line 27. Recommend to be more specific in the summary statement: "...mice, in the context of a nonconditioned transplantation model, to assess simultaneously...".
2. Line 70. Although a lower frequency, Myeloid/MK-HSCs can be found within CD41- gated HSCs – "reduce" may be a more correct term than "avoid"
3. Figure 1A-C. While it is useful to have the side-by-side comparison of pre-and post-transplanted cell FACS data, it would be even easier to compare if (1) the FACS plots had the same x and y axes, and (2) the same Sca1 gating was used (it appears a higher threshold is used for the post-transplantation gating).
4. Related to the author response to my initial comment 3. It is unfortunate that IF was not possible for these experiments. In lieu, it may be relevant to discuss CFSE-labelling experiments by Takizawa et al JEM 2011, which suggest three-weeks after non-conditioned transplantation ("steady-state" conditions), non-dividing LSK cells uniformly retain high Sca1 expression. It is important whether the described HSC-MP non-dividing differentiation route is active directly following transplantation stress or in more steady-state conditions.
5. Figure 3. Given the resolution/separation of these populations are not so clear, this may be more appropriate as a supplementary figure.
6. Line 157. "transplanted HSCs can differentiate" would be more accurate.
7. Line 170. "that phenotypic HSCs do not..." would be more accurate.
8. Figure 5A and related to my initial comment 17. The authors state HSCs and PreMeg potency is similar but it appears different criteria for HSC and PreMeg colonies were used. It would be more accurate to use the same criteria for this comparison (e.g. it might increase HSC potency to include any colony of 3 or more cells).

9. Figure 5A – unclear what “Total=99.9” refers to. Suggest deletion.
10. Figure 5C – from looking at 5D, it would seem these plots are not “representative” as described in the legend, but the “champion” data. While the data don’t necessarily need to be changed, “example plots” might be a better term than “representative plots”. Legend for 5D should also note that these data are from 3w PB analysis.
11. Figure 5B-D and related to my initial comment 18. Given 16 week PB analysis was performed, it would be best to present this data. It would be encouraging to see that the LSK retain long-term functional HSC capacity (and lymphoid capacity), while (as stated by the authors) MP lack this functional capacity. If neither contributes long-term (or to lymphoid lineages), it would suggest that the non-conditioned transplantation is causing some sort of stress that is perturbing HSC function (and neither population contains functional HSCs). If such a caveat exists, it should at minimum be discussed.
12. Delete or reword sentence starting line 220. There is no evidence that the same mechanism is responsible for both HSC and ESC fate decisions. Similar revision suggested for line 40.
13. Line 396 – the authors should better define the “HSC-CreERT” mouse. Ages of mice used should also be included in the methods.

Reviewer #2 (Remarks to the Author):

The authors have addressed most previous concerns. However, the strength of their previous and new data as well as their rebuttals do not make a strong case for the very dogmatic statements such as that carried in the title, line 34, line 64, etc. The data are credibly consistent with the idea that HSCs can restrict their differentiation options before they enter S-phase but they do not prove that this is the only way restriction can occur - otherwise it would never be possible to detect multipotent HSCs at nearly 100% efficiency. There is also a tendency to reference studies of human and mouse HSC regulation as if these mechanisms are completely the same, which clearly they are not. Therefore, my only remaining comment would be to tone down the definitive tone of the conclusions and recognize that the findings need further exploration for their applicability to human HSCs.

Reviewer #1 (Remarks to the Author):

The authors have clearly gone to considerable effort to revising this manuscript, and their comprehensive responses are highly appreciated by this reviewer. The manuscript and figures are now easier to follow and understand. The paper suggests an interesting concept for the HSC field that deserves publication, although definitive evidence using heterogeneous HSC populations is challenging, and physiological relevance of this mechanism remains unclear.

We thank the reviewer for these very stimulating words.

The following reservations remain:

1. *Line 27. Recommend to be more specific in the summary statement: "...mice, in the context of a nonconditioned transplantation model, to assess simultaneously..."*

We thank the reviewer for the suggestion and have now changed the summary statement.

2. *Line 70. Although a lower frequency, Myeloid/MK-HSCs can be found within CD41- gated HSCs – "reduce" may be a more correct term than "avoid"*

We agree with the reviewer and changed the word 'avoid' into 'reduce'

3. *Figure 1A-C. While it is useful to have the side-by-side comparison of pre-and post-transplanted cell FACS data, it would be even easier to compare if (1) the FACS plots had the same x and y axes, and (2) the same Sca1 gating was used (it appears a higher threshold is used for the post-transplantation gating).*

In Figure 1a-c, we have now (1) changed the order of the axes, as suggested by the reviewer. With regard to the issue on Sca1 (2): we have not been able to use the same gating, since the sort + re-analysis versus donor cell analysis (=36h after transplantation) are obviously performed on different days. However, we have always used recipient cells in both approaches that served as control for gating of LSK cells.

4. *Related to the author response to my initial comment 3. It is unfortunate that IF was not possible for these experiments. In lieu, it may be relevant to discuss CFSE-labelling experiments by Takizawa et al JEM 2011, which suggest three-weeks after non-conditioned transplantation ("steady-state" conditions), non-dividing LSK cells uniformly retain high Sca1 expression. It is important whether the described HSC-MP non-dividing differentiation route is active directly following transplantation stress or in more steady-state conditions.*

This is an interesting remark, which we now discussed in the discussion-section of our revised manuscript. Even though no direct *in vivo* information is available in the literature, it is very well possible that the "HSC-MP non-dividing differentiation route" is active directly following transplantation stress. Additionally, it might also be easier to detect such events, as during steady state conditions only a small fraction of HSCs are non-quiescent and therefore can only produce a limited amount of differentiated progenitors.

Nevertheless, the key finding of our work is the capability of HSCs to decide to differentiate into a downstream progenitor without physical division, even before progressing into the S phase of the cell cycle. Our findings therefore suggest that differentiation and division of an HSC can exist as two independent processes.

5. *Figure 3. Given the resolution/separation of these populations are not so clear, this may be more appropriate as a supplementary figure.*

We have assessed the concern of the reviewer and decided to move Figure 3a into the supplementary figures (new Supplementary Figure 5e). Because of the impact of Figure 3b-c for our entire story, we decided to keep these two graphs and renamed them to Figure 3a-b.

6. *Line 157. “transplanted HSCs can differentiate” would be more accurate.*

We changed it in the text.

7. *Line 170. “that phenotypic HSCs do not...” would be more accurate.*

We changed it in the text.

8. *Figure 5A and related to my initial comment 17. The authors state HSCs and PreMeg potency is similar but it appears different criteria for HSC and PreMeg colonies were used. It would be more accurate to use the same criteria for this comparison (e.g. it might increase HSC potency to include any colony of 3 or more cells).*

We understand the concern of the reviewer. However, as the aim of this approach was to define the composition of the different colonies from the same cell type (either HSC or preMeg), the criteria between the cell types do not necessarily have to be the same. Actually, these experiments required two different approaches. Firstly, we used a 20-cell criterion for **HSCs** since the colony composition can only be convincingly analyzed from this size. Secondly, for the colony assay with **preMegs** it is much more feasible to detect giant megakaryocytes, and therefore a 3-cell criterion was sufficient to undoubtedly define the composition of the individual colonies.

9. *Figure 5A – unclear what “Total=99.9” refers to. Suggest deletion.*

We deleted it.

10. *Figure 5C – from looking at 5D, it would seem these plots are not “representative” as described in the legend, but the “champion” data. While the data don’t necessarily need to be changed, “example plots” might be a better term than “representative plots”. Legend for 5D should also note that these data are from 3w PB analysis.*

We changed the figure legend accordingly.

11. *Figure 5B-D and related to my initial comment 18. Given 16 week PB analysis was performed, it would be best to present this data. It would be encouraging to see that the LSK retain long-term functional HSC capacity (and lymphoid capacity), while (as stated by the authors) MP lack this functional capacity. If neither contributes long-term (or to lymphoid lineages), it would suggest that the non-conditioned transplantation is causing some sort of stress that is perturbing HSC function (and neither population contains functional HSCs). If such a caveat exists, it should at minimum be discussed.*

As suggested by the reviewer, we have now added the peripheral blood analysis after 16 weeks into the supplementary figures (Supplementary Figure 6d).

12. *Delete or reword sentence starting line 220. There is no evidence that the same mechanism is responsible for both HSC and ESC fate decisions. Similar revision suggested for line 40.*

We reformulated line 220 and deleted line 40.

13. *Line 396 – the authors should better define the “HSC-CreERT” mouse. Ages of mice used should also be included in the methods.*

We have now added more information on this mouse line into the methods section.

Reviewer #2 (Remarks to the Author):

The authors have addressed most previous concerns. However, the strength of their previous and new data as well as their rebuttals do not make a strong case for the very dogmatic statements such as that carried in the title, line 34, line 64, etc. The data are credibly consistent with the idea that HSCs can restrict their differentiation options before they enter S-phase but they do not prove that this is the only way restriction can occur - otherwise it would never be possible to detect multipotent HSCs at nearly 100% efficiency. There is also a tendency to reference studies of human and mouse HSC regulation as if these mechanisms are completely the same, which clearly they are not. Therefore, my only remaining comment would be to tone down the definitive tone of the conclusions and recognize that the findings need further exploration for their applicability to human HSCs.

We thank the reviewer for the positive comments and the fact that he/she finds our data credibly consistent with our key finding that HSCs can decide *in vivo* to differentiate into a downstream progenitor without physical division. Our findings also suggest that differentiation and division of an HSC can therefore exist as two independent processes. However, we have never had the intention to propose that our data demonstrate the only possible way restriction can occur. On the contrary, we are convinced that division-related differentiation is an existing route, which will depend on the cell type, its environment and/or activation state. We have carefully checked our manuscript again and have corrected any potential insinuation in that direction.

As suggested by this reviewer, we have also toned down the conclusion in the different sections, including the title, and added that our findings need further exploration for their applicability to human HSCs.